# STay-ON-the-Ridge: Guaranteed Convergence to Local Minimax Equilibrium in Nonconvex-Nonconcave Games

## Abstract

Min-max optimization problems involving nonconvex-nonconcave objectives have found important applications in adversarial training and other multi-agent learning settings. Yet, no known gradient descent-based method is guaranteed to converge to (even local notions of) min-max equilibrium in the nonconvex-nonconcave setting. For all known methods, there exist relatively simple objectives for which they cycle or exhibit other undesirable behavior different from converging to a point, let alone to some game-theoretically meaningful one Vlatakis-Gkaragkounis et al. (2019); Hsieh et al. (2021). The only known convergence guarantees hold under the strong assumption that the initialization is very close to a local min-max equilibrium Wang et al. (2019). Moreover, the afore-described challenges are not just theoretical curiosities. All known methods are unstable in practice, even in simple settings.

We propose the first method that is guaranteed to converge to a local min-max equilibrium for smooth nonconvex-nonconcave objectives. Our method is second-order and provably escapes limit cycles as long as it is initialized at an easy-to-find initial point. Both the definition of our method and its convergence analysis are motivated by the topological nature of the problem. In particular, our method is not designed to decrease some potential function, such as the distance of its iterate from the set of local min-max equilibria or the projected gradient of the objective, but is designed to satisfy a topological property that guarantees the avoidance of cycles and implies its convergence.

## 1 Introduction

Min-max optimization lies at the foundations of Game Theory von Neumann (1928), Convex Optimization Dantzig (1951a); Adler (2013) and Online Learning Blackwell (1956); Hannan (1957); Cesa-Bianchi & Lugosi (2006), and has found many applications in theoretical and applied fields including, more recently, in adversarial training and other multi-agent learning problems Goodfellow et al. (2014); Madry et al. (2018); Zhang et al. (2019). In its general form, it can be written as

$$\min_{\theta \in \Theta} \max_{\omega \in \Omega} f(\theta, \omega), \tag{1}$$

where $\Theta$ and $\Omega$ are convex subsets of the Euclidean space, and $f$ is continuous.

Equation (1) can be viewed as a model of a sequential-move game wherein a player who is interested in minimizing $f$ chooses $\theta$ first, and then a player who is interested in maximizing $f$ chooses $\omega$ after seeing $\theta$. Solving (1) corresponds to an equilibrium of this sequential-move game.

We may also study the simultaneous-move game with the same objective $f$ wherein the minimizing player and the maximizing player choose $\theta$ and $\omega$ simultaneously. The Nash equilibrium of the simultaneous-move game, also called a *min-max equilibrium*, is a pair $(\theta^\star, \omega^\star) \in \Theta \times \Omega$ such that

$$f(\theta^\star, \omega^\star) \leq f(\theta, \omega^\star), \text{ for all } \theta \in \Theta \text{ and } f(\theta^\star, \omega^\star) \geq f(\theta^\star, \omega), \text{ for all } \omega \in \Omega. \tag{2}$$

It is easy to see that a Nash equilibrium of the simultaneous-move game also constitutes a Nash equilibrium of the sequential-move game, but the converse need not be true Jin et al. (2019). Here, we focus on solving the (harder) simultaneous-move game. In particular, we study the existence

of *dynamics* which converge to solutions of the simultaneous-move game, namely the existence of methods that make incremental updates to a pair $(\theta_t, \omega_t)$ so as the sequence $(\theta_t, \omega_t)$ converges, as $t \to \infty$, to some $(\theta^*, \omega^*)$ satisfying equation 2 or some relaxation of it.

This problem has been extensively studied in the special case where $\Theta$ and $\Omega$ are convex and compact and $f$ is convex-concave — i.e. convex in $\theta$ for all $\omega$ and concave in $\omega$ for all $\theta$. In this case, the set of Nash equilibria of the simultaneous-move game is equal to the set of Nash equilibria of the sequential-move game, and these sets are non-empty and convex von Neumann (1928). Even in this simple setting, however, many natural dynamics surprisingly fail to converge: *gradient descent-ascent*, as well as various continuous-time versions of *follow-the-regularized-leader*, not only fail to converge to a min-max equilibrium, even for very simple objectives, but may even exhibit chaotic behavior Mertikopoulos et al. (2018); Vlatakis-Gkaragkounis et al. (2019); Hsieh et al. (2021). In order to circumvent these negative results, an extensive line of work has introduced other algorithms, such as *extragradient* Korpelevich (1976) and *optimistic gradient descent* Popov (1980), which exhibit last-iterate convergence to the set of min-max equilibria in this setting; see e.g. Daskalakis et al. (2018); Daskalakis & Panageas (2018); Mazumdar & Ratliff (2018); Rafique et al. (2018); Hamedani & Aybat (2018); Adolphs et al. (2019); Daskalakis & Panageas (2019); Liang & Stokes (2019); Gidel et al. (2019); Mokhtari et al. (2019); Abernethy et al. (2019); Golowich et al. (2020b;a); Gorbunov et al. (2022); Cai et al.. Alternatively, one may take advantage of the convexity of the problem, which implies that several no-regret learning procedures, such as online gradient descent, exhibit *average*-iterate convergence to the set of min-max equilibria Cesa-Bianchi & Lugosi (2006); Shalev-Shwartz (2012); Bubeck & Cesa-Bianchi (2012); Shalev-Shwartz & Ben-David (2014); Hazan (2016). Beyond the convex/concave setting Lin et al. (2020); Kong & Monteiro (2021); Ostrovskii et al. (2021) show that convexity with respect to one of the two players is enough to design algorithms that exhibit average-iterate convergence to min-max equilibria while Diakonikolas et al. (2021) and Pethick et al. (2022) provide convergence results for *weak Minty variational inequalities*.

Our focus in this paper is on the more general case where $f$ is not convex-concave, i.e. it may fail to be convex in $\theta$ for all $\omega$, or may fail to be concave in $\omega$ for all $\theta$, or both. We call this general setting where neither convexity with respect to $\theta$ nor concavity with respect to $\omega$ is assumed, the *nonconvex-nonconcave* setting. This setting presents some substantial challenges. First, min-max equilibria are *not* guaranteed to exist, i.e. for general objectives there may be no $(\theta^*, \omega^*)$ satisfying equation 2; this happens even in very simple cases, e.g. when $\Theta = \Omega = [0, 1]$ and $f(\theta, \omega) = (\theta - \omega)^2$. Second, it is NP-hard to determine whether a min-max equilibrium exists Daskalakis et al. (2021) and, as is easy to see, it is also NP-hard to compute Nash equilibria of the sequential-move game (which do exist under compactness of the constraint sets). For these reasons, the optimization literature has targeted the computation of local and/or approximate solutions in this setting Daskalakis & Panageas (2018); Mazumdar & Ratliff (2018); Jin et al. (2019); Wang et al. (2019); Daskalakis et al. (2021); Mangoubi & Vishnoi (2021). This is the approach we also take in this paper, targeting the computation of $(\epsilon, \delta)$-*local min-max equilibria*, which were proposed in Daskalakis et al. (2021). These are approximate and local Nash equilibria of the simultaneous-move game, defined as feasible points $(\theta^*, \omega^*)$ which satisfy a relaxed and local version of equation 2, namely:

$$f(\theta^*, \omega^*) < f(\theta, \omega^*) + \epsilon, \quad \text{for all } \theta \in \Theta \text{ such that } \|\theta - \theta^*\| \leq \delta; \tag{3}$$

$$f(\theta^*, \omega^*) > f(\theta^*, \omega) - \epsilon, \quad \text{for all } \omega \in \Omega \text{ such that } \|\omega - \omega^*\| \leq \delta. \tag{4}$$

Besides being a natural concept of local, approximate min-max equilibrium, an attractive feature of $(\epsilon, \delta)$-local min-max equilibria is that they are guaranteed to exist when $f$ is $\Lambda$-smooth and the locality parameter, $\delta$, is chosen small enough in terms of the smoothness, $\Lambda$, and the approximation parameter, $\epsilon$, namely whenever $\delta \leq \sqrt{\frac{2\epsilon}{\Lambda}}$. Indeed, in this regime of parameters the $(\epsilon, \delta)$-local min-max equilibria are in correspondence with the approximate fixed points of the *Projected Gradient Descent/Ascent* dynamics. Thus, the existence of the former can be established by invoking Brouwer's fixed point theorem to establish the existence of the latter. (Theorem 5.1 of Daskalakis et al. (2020)).

There are a number of existing approaches which would be natural to use to find a solution $(\theta^*, \omega^*)$ satisfying equation 3 and equation 4, but all run into significant obstacles. First, the idea of averaging, which can be leveraged in the convex-concave setting to obtain provable guarantees for otherwise chaotic algorithms, such as online gradient descent, no longer works, as it critically uses Jensen's inequality which needs convexity/concavity. On the other hand, negative results abound for last-iterate convergence: Hsieh et al. (2021) show that a variety of zeroth, first, and second order methods may converge to a limit cycle, even in simple settings. Vlatakis-Gkaragkounis et al. (2019) study a

particular class of nonconvex-nonconcave games and show that continuous-time gradient descent-ascent (GDA) exhibits *recurrent* behavior. Furthermore, common variants of gradient descent-ascent, such as optmistic GDA (OGDA) or extra-gradient (EG), may be unstable even in the proximity of local min-max equilibria, or converge to fixed points that are not local min-max equilibria Daskalakis & Panageas (2018); Jin et al. (2019). While there do exist algorithms, such as FOLLOW-THE-RIDGE proposed by Wang et al. (2019), which provably exhibit *local convergence* to a (relaxation of) local min-max equilibrium, these algorithms do not enjoy global convergence guarantees, and no algorithm is known with guaranteed convergence to a local min-max equilibrium.

These negative theoretical results are consistent with the practical experience with min-maximization of nonconvex-nonconcave objectives, which is rife with frustration as well. A common experience is that the training dynamics of first-order methods are unstable, oscillatory or divergent, and the quality of the points encountered in the course of training can be poor; see e.g. Goodfellow (2016); Metz et al. (2016); Daskalakis et al. (2018); Mescheder et al. (2018); Daskalakis & Panageas (2018); Mazumdar & Ratliff (2018); Mertikopoulos et al. (2018); Adolphs et al. (2019). In light of the failure of essentially all non-trivial, i.e., non brute-force, algorithms to guarantee convergence, even asymptotically, to local min-max equilibria, we ask the following question: *Is there any local-search algorithm which is guaranteed to converge to a local min-max equilibrium in the nonconvex-nonconcave setting? (see Table 1)*

## 1.1 OUR CONTRIBUTION

In this work we answer the above question in the affirmative: **we propose a second-order method that is guaranteed to converge to a local min-max equilibrium (Theorem 1).**. Our algorithm, called STAY-ON-THE-RIDGE or STON'R, has some similarity to FOLLOW-THE-RIDGE or FTR, which only converges locally. STON'R is the first method guaranteed to local min-max equilibrium beyond the brute-force grid-search in the non-convex/non-concave setting. Both the structure of our algorithm and its global convergence analysis are motivated by the topological nature of the problem, as established by Daskalakis et al. (2021) who showed that the problem is equivalent to Brouwer fixed point computation. In particular, the structure and analysis of STON'R are not based on a potential function argument but on a *parity argument* (see Section 4), akin to the argument used to prove the existence of Brouwer fixed points. The main challenge of our work is to prove that there exists an algorithm that uses only *local information* of the objective function $f$, i.e., only its second derivative, while satisfying the topological properties that are necessary to guarantee global convergence. In order to understand the main technical contributions of our paper we need first to introduce the main steps of showing the convergence using a topological argument in Section 4. Then in Section 5.4 we provide a sketch of our proof and we highlight the technical difficulties that we face.

| | | convex-concave | nonconvex-concave | **nonconvex-nonconcave** |
|---|---|---|---|---|
| **Nash Eq.** | existence | **yes**[□] | **no**[†] | **no**[†] |
| | complexity | **poly-time**[‡] | **NP-hard**[⋆] | **NP-hard**[⋆] |
| | convergent dynamics | **many**[‡] | not applicable | not applicable |
| **Local Nash Eq.** | existence | *same as above* | **yes**[+] | **yes**[⋆] |
| | complexity | *same as above* | **poly-time**[+] | **PPAD-hard**[⋆] |
| | convergent dynamics | *same as above* | **many**[+] | **This paper** |

Table 1: Summary of known results for simultaneous zero-sum games with differing complexity in their objective function. (□) v. Neumann (1928) (†) e.g., the min-max game with objective function $f(\theta, \omega) = -(\theta - \omega)^2$, where $\theta \in [-1, 1]$ and $\omega \in [-1, 1]$, does not have any Nash Equilibrium. (⋆) Daskalakis et al. (2021) (‡) e.g., Dantzig (1951b); Freund & Schapire (1997); Shalev-Shwartz (2012); Cesa-Bianchi & Lugosi (2006) (+) e.g., Lin et al. (2020); Kong & Monteiro (2021); Ostrovskii et al. (2021)

## 2 SOLUTION CONCEPT

We begin with formulating our problem in the more general framework of *variational inequalities*. This simplifies our definitions and notations and also makes our result applicable to more general settings such as multi-player concave games Rosen (1965).

**Variational Inequalities (VI).** For $K \subseteq \mathbb{R}^n$, consider a continuous map $V : K \to \mathbb{R}^n$. We say that $x \in K$ is a solution of the variational inequality VI$(V, K)$ iff: $V(x)^\top \cdot (x - y) \geq 0$ for all $y \in K$.

It is well known that finding local min-max equilibria of smooth objectives can be expressed as a non-monotone VI problem. Specifically, consider the min-max optimization problem (1), take $K = \Theta \times \Omega$ and simplify notation by using $x \in K$ to denote points $(\theta, \omega) \in K$. Call the subset of coordinates of $x$ identified with $\theta$ the "*minimizing* coordinates" and the subset of coordinates of $x$ identified with $\omega$ the "*maximizing* coordinates." Then define $V : K \to \mathbb{R}^n$ as follows:

$$\text{For } j \in [n]: \text{set } V_j(x) := -\frac{\partial f(x)}{\partial x_j}, \text{ if } j \text{ is minimizing, and } V_j(x) := \frac{\partial f(x)}{\partial x_j}, \text{ otherwise.}$$

Computing $(\varepsilon, \delta)$-local min-max equilibria of smooth objectives, i.e. points satisfying (3) & (4), can be reduced to finding solutions to VI$(V, K)$. In fact, finding even an approximate VI solution $x$ satisfying $V(x)^\top (x - y) \geq -\alpha, \forall y \in K$, would suffice as long as $\alpha > 0$ is small enough. For more details see Theorem 5.1 of Daskalakis et al. (2020). Hence, for the rest of the paper we focus on solving variational inequality problems. For simplicity of exposition we take our constraint set to be $K = [0, 1]^n$. In this case there is a simple characterization of the solutions to VI$(V, K)$.

**Definition 1.** *We call a coordinate $i$ **satisfied** at point $x \in [0, 1]^n$ if one of the following holds:*

1. *$i$ is **zero-satisfied** at $x$, i.e, $V_i(x) = 0$, or*

2. *$i$ is **boundary-satisfied** at $x$, i.e, $(V_i(x) \leq 0$ and $x_i = 0)$ or $(V_i(x) \geq 0$ and $x_i = 1)$.*

**Lemma 1** (Proof in Appendix D). *$x$ is a solution of VI$(V, [0, 1]^n)$ iff $j$ is satisfied at $x, \forall j \in [n]$.*

Finally, in the rest of the paper we make the following assumptions for $V$:

> **($\Lambda$-Lipschitz)**   $\|V(x) - V(y)\|_2 \leq \Lambda \cdot \|x - y\|_2$, for all $x, y \in [0, 1]^n$.
> **(*L*-smooth)**   $\|J(x) - J(y)\|_F \leq L \cdot \|x - y\|_2$, for all $x, y \in [0, 1]^n$.

where $J$ is the Jacobian of V, and $\|A\|_F$ denotes the Frobenious norm of the matrix $A$.

## 3 STAY-ON-THE-RIDGE: HIGH-LEVEL DESCRIPTION

In this section we describe our algorithm and discuss the main design ideas leading to its convergence properties presented in Section 5. As explained in the previous section, our goal is to find a point $x$ such that every coordinate $i \in [n]$ is satisfied at $x$ according to the Definition 1.

Our algorithm is initialized at $x(0) = (0, \ldots, 0)$. The goal of the algorithm is to satisfy all unsatisfied coordinates one-by-one in lexicographic order (although, as we will see, coordinates may go from being satisfied to being unsatisfied in the course of the algorithm). We say that our algorithm "starts epoch $i$ at point $x$" iff all coordinates $\leq i - 1$ are satisfied at $x$ and the algorithm's immediate goal is to find a point $x' \neq x$ that satisfies all coordinates $\leq i$, namely:

> Goal of epoch $i$, starting at point $x$: find $x' \neq x$ satisfying all coordinates $\leq i$.

Let us assume that, at time $t$, our algorithm starts epoch $i$ at point $x(t)$. Let us also assume that, at $x(t)$, all coordinates $\leq i - 1$ are zero-satisfied (see Section 5.1 for the general case), i.e., $V_j(x(t)) = 0$ for all $j \leq i - 1$. Our algorithm tries to achieve the goal of epoch $i$ starting at $x(t)$ as follows:

- Our algorithm tries to find such a point inside the connected subset $S^i(x(t)) \subseteq [0, 1]^n$ that contains all points $z$ satisfying the following: (a) all coordinates $\leq i - 1$ are zero-satisfied at $z$, and (b) for all $j \geq i + 1$, $z_j = x_j(t)$.

- Our algorithm navigates $S^i(x(t))$ in the hopes of satisfying the goal of epoch $i$. A natural approach is to navigate $S^i(x(t))$ is to run a continuous-time dynamics $\{z(\tau)\}_{\tau \geq 0}$ that is initialized at $z(0) = x(t)$ and moves inside $S^i(x(t))$. What are possible directions of movement so that our dynamics stay within $S^i(x(t))$? If the dynamics is at some point $z \in S^i(x(t))$, it will remain in this set if it moves, infinitessimally, in a unit direction $d$ satisfying the following constraints:

1. $d_j = 0$, for all $j \geq i + 1$;   /* to guarantee (b) in the definition of $S^i(x(t))$ */
2. $(\nabla V_j(z))^\top \cdot d = 0$, for all $j \in \{1, \ldots, j-1\}$.   /* to guarantee (a) */

Notice that 1 and 2 specify $n-1$ constraints on $n$ variables. We will place mild assumptions on $V$ so that there is a unique, up to a sign flip, unit direction satisfying these constraints (see Assumption 1). Moreover, in Definition 2 we specify a rule to choose one of the two unit directions satisfying our constraints. We denote by $D^i(z)$ the direction that our tie-breaking rule selects at $z$.

- With the above choices, the continuous-time dynamics $\dot{z}(\tau) = D^i(z(\tau))$, initialized at $z(0) = x(t)$, is well-defined. We follow this dynamics until the earliest time that one of the following happens:
  - (Good Event): the dynamics stops at a point $x' \neq x(t)$ where coordinate $i$ is satisfied;
  - (Bad Event): the dynamics stops at a point $x'$ lying on the boundary of $[0,1]^n$ (and if it were to continue it would violate the constraints).

So we have described what our algorithm does if, at time $t$, it starts epoch $i$ at $x(t)$. Suppose $x'$ is the point where our dynamics executed during epoch $i$ terminates. If the good event happened, coordinate $i$ is satisfied at $x'$, and our algorithm starts epoch $i+1$ at $x'$. If the bad event happened, our algorithm will in fact *start epoch $i-1$* at point $x'$. What does this mean? That it will run the continuous-time dynamics corresponding to epoch $i-1$ on the set $S^{i-1}(x')$ starting at $x'$ in order to find some point $x'' \neq x'$ where all coordinates $\leq i-1$ are satisfied. It may fail to do this, in which case it will start epoch $i-2$ next. Or it may succeed, in which case, it will start epoch $i$, and so on so forth until (as we will show!) all coordinates will be satisfied. The high-level pseudocode of our algorithm is given in Dynamics 1.

---

**Dynamics 1** STay-ON-the-Ridge (STON'R) — High-Level Description

1: Initially $x(0) \leftarrow (0, \ldots, 0)$, $i \leftarrow 1$, $t \leftarrow 0$.
2: **while** $x(t)$ is not a VI solution **do**
3:     Initialize epoch $i$'s continuous-time dynamics, $\dot{z}(\tau) = D^i(z(\tau))$, at $z(0) = x(t)$.
4:     **while** exit condition of this dynamics has not been reached **do**
5:         Execute $\dot{z}(\tau) = D^i(z(\tau))$ forward in time.
6:     **end while**
7:     Set $x(t + \tau) = z(\tau)$ for all $\tau \in [0, \tau_{\text{exit}}]$ (where $\tau_{\text{exit}}$ is time exit condition was met).
8:     **if** $x(t + \tau_{\text{exit}}) \neq x(t)$ and coordinate $i$ is satisfied at $x(t + \tau_{\text{exit}})$ **then**
9:         Update the epoch $i \leftarrow i + 1$.
10:    **else**
11:        (Bad event happened so) move to the previous epoch $i \leftarrow i - 1$.
12:    **end if**
13:    Set $t \leftarrow t + \tau_{\text{exit}}$.
14: **end while**
15: **return** $x(t)$

---

At this point we have described an algorithm that explores the space in a natural way in its effort to satisfy coordinates, but it is unclear why it would eventually satisfy all of them, how it would escape cycles, and how it would not get stuck at non-equilibrium points. Importantly, there is no quantity that seems to be consistently improving during the execution of the algorithm.

*How we can show convergence since no quantity seems to be consistently improving?*

## 4   A TOPOLOGICAL ARGUMENT OF CONVERGENCE

Our main idea to show the convergence of the STON'R algorithm is to use a topological argument illustrated in Lemma 2 that has been employed to show the convergence of other equilibrium computation algorithms such as the elebrated Lemke-Howson algorithm Lemke & Howson (1964).

**Lemma 2.** *Let $G = (N, E)$ be a directed graph such that every node has in-degree at most $1$ and out-degree at most $1$. If there exists some node $v \in N$ with in-degree $0$ and out-degree $1$, then there is unique directed path starting at $v$ and ending at some $v' \in N$ that has in-degree $1$ and out-degree $0$.*

The proof of Lemma 2 is straightforward, as Figure 1 illustrates. The lemma suggests a recipe for proving the convergence of some deterministic, iterative algorithm, with update rule $v_{t+1} \leftarrow F(v_t)$, whose iterates lie in a finite set $N$:

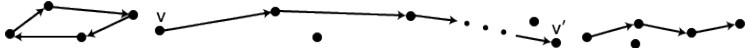

Figure 1: A directed graph whose nodes have in-degree and out-degree at most 1 is a collection of directed paths, directed cycles, and isolated nodes. Hence, if a node $v$ has in-degree 0 and out-degree 1 then it has to be the start of a directed path that must end at a node $v'$ after a finite number of steps.

1. Define a graph $G$ with vertices $N$ and edges $E = \{(u, v) \mid u \neq v \text{ and } v = F(u)\}$, i.e., there is an edge from $u$ to $v$ iff $v \neq u$ and $v$ is reached after an iteration of the algorithm starting at $u$.
2. Argue that every vertex of $G$ has in-degree $\leq 1$. It is clear that every vertex has out-degree $\leq 1$.
3. Show that the algorithm can be initialized at some $v_0$ that has in-degree 0 and out-degree 1.
4. Employ Lemma 2 to argue that if the algorithm is initialized at $v_0$ it must, eventually, arrive at some node $v_{\text{end}}$ whose out-degree is 0. Out-degree 0 means that $v_{\text{end}} = F(v_{\text{end}})$.
5. The above prove that if the algorithm starts at $v_0$ it is guaranteed to converge.

In the course of the description of the algorithm and its convergence proof in Section 5, we specify a finite set of nodes $N$ of the graph that we will construct to employ the above convergence argument. Intuitively, these are all the points at which our algorithm can possibly start a new epoch. The map $F(\cdot)$ that we use to construct our graph is the outcome of the continuous-time process that our algorithm execute when it starts an epoch at such a point.

## 5 DETAILED DESCRIPTION OF STON'R AND MAIN RESULT

We provide a formal description of our algorithm (Section 5.1), state our main convergence theorem (Section 5.2), and the main components of its proof building on the ideas (Section 5.4).

### 5.1 STON'R: DETAILED DESCRIPTION

In Section 3 we focused on the epochs where all coordinates $\leq i - 1$ are zero-satisfied at the initial point $x$ and the goal is to identify some $x' \neq x$ all coordinates $\leq i$ are satisfied. To achieve this, we execute a continuous-time dynamics constrained by keeping all coordinates $\leq i - 1$ zero-satisfied. However, in the course of these dynamics we be hit the boundary. So, when we start a new epoch, some coordinates will be zero-satisfied and some will be boundary-satisfied. In that general case, the algorithm needs to execute a continuous-time dynamics constrained by keeping the zero-satisfied coordinates zero-satisfied as well as the boundary-satisfied coordinates at the right boundary.

Namely, the epochs are indexed by some coordinate $i \in [n]$ and a subset of coordinates $S \subseteq [i - 1]$ that are zero-satisfied at the point $x$ where the epoch starts. The goal of each epoch is the following.

*Goal of epoch $(i, S)$, starting at point $x$ (where $S \subseteq [i - 1]$, coordinates in $S$ are zero-satisfied and coordinates in $[i - 1] \setminus S$ are boundary-satisfied): find $x' \neq x$ where all coordinates $\leq i$ are satisfied, all coordinates in $S$ are zero-satisfied, and all coordinates in $[i - 1] \setminus S$ are boundary-satisfied.*

Epoch $(i, S)$ starting at $x$ might achieve its goal or end before it achieves its goal. In both cases, a new epoch will start. Within each epoch our algorithm executes a continuous-time dynamics that maintains all the coordinates $j \in S$ zero-satisfied, all the coordinates $j \in [i - 1] \setminus S$ boundary-satisfied, and leaves all coordinates $[n] \setminus [i]$ unchanged.

**Definition 2** (Tangent Unit Vector of Epoch $(i, S)$). *Let $i \in [n]$, $S = \{s_1, \ldots, s_m\} \subseteq [i - 1]$, and $x \in [0, 1]^n$, we say that a unit vector $d \in \mathbb{R}^n$ is admissible if:*

1. *$d_j = 0$, for all $j \notin S \cup \{i\}$, and*

2. *$\nabla V_j(x)^\top \cdot d = 0$, for all $j \in S$, and*

3. *the sign of*
$$\begin{vmatrix} \frac{\partial V_{s_1}(x)}{\partial x_{s_1}} & \frac{\partial V_{s_2}(x)}{\partial x_{s_1}} & \cdots & \frac{\partial V_{s_m}(x)}{\partial x_{s_1}} & d_{s_1} \\ \vdots & \vdots & \vdots & \vdots & \vdots \\ \frac{\partial V_{s_1}(x)}{\partial x_{s_m}} & \frac{\partial V_{s_2}(x)}{\partial x_{s_m}} & \cdots & \frac{\partial V_{s_m}(x)}{\partial x_{s_m}} & d_{s_m} \\ \frac{\partial V_{s_1}(x)}{\partial x_i} & \frac{\partial V_{s_2}(x)}{\partial x_i} & \cdots & \frac{\partial V_{s_m}(x)}{\partial x_i} & d_i \end{vmatrix}$$
*equals the sign of $(-1)^{|S|}$.*

*If there is a unique unit direction satisfying the above constraints, we denote that direction $D_S^i(x)$.*

Conditions 1 and 2 above describe a line in $\mathbb{R}^n$ and condition 3 specifies a direction on this line. We will place some assumptions on $V$ so that $D_S^i(x)$ is defined for all $x \in [0,1]^n$ where coordinates $S$ are zero-satisfied (see Assumption 1). Now, when we start epoch $(i, S)$ at point $x$, we will execute the continuous-time dynamics $\dot{z}(\tau) = D_S^i(z(\tau))$, initialized at $z(0) = x$, forward in time. We this dynamics until the earliest time $\tau_{\mathrm{exit}}$ such that $z(\tau_{\mathrm{exit}})$ is an *exit point* according to the next definition.

**Definition 3.** *Suppose $i \in [n]$, $S \subseteq [i-1]$, at $x' \in [0,1]^n$, the coordinates in $S$ are zero-satisfied at $x'$, the coordinates in $[i-1] \setminus S$ are boundary-satisfied at $x'$. Then $x'$ is an* exit point *for epoch $(i, S)$ iff it satisfies one of the following:*

- *(**Good Exit Point**): Coordinate $i$ is satisfied at $x'$, i.e., $V_i(x') = 0$, or $x_i' = 0$ and $V_i(x') < 0$, or $x_i' = 1$ and $V_i(x') > 0$.*
- *(**Bad Exit Point**): $\exists j \in S \cup \{i\}$ s.t. $(D_S^i(x'))_j > 0$ and $x_j' = 1$, or $(D_S^i(x'))_j < 0$ and $x_j' = 0$, i.e., if the dynamics of epoch $(i, S)$ were to continue from $x'$, they would violate the constraints.*
- *(**Middling Exit Point**): $\exists j \in [i-1] \setminus S$ s.t. $V_j(x') = 0$ and $(\nabla V_j(x')^\top D_S^i(x') > 0$ and $x_j' = 0)$ or $(\nabla V_j(x')^\top D_S^i(x') < 0$ and $x_j' = 1)$, i.e., if the dynamics for epoch $(i, S)$ were to continue from $x'$, some boundary-satisfied coordinate would become unsatisfied.*

We will place some assumptions on $V$ so that there can be a unique $j$ triggering the condition of Bad Exit Point and there can be a unique $j$ triggering the Middling Exit Point condition (see Assumptions 2). Below we describe the actions that we take when one of the above exit conditions is triggered.

**Action at Good Events.** In case of a good event, we start epoch $(i+1, S')$ at $x'$, where $S' = S \cup \{i\}$, if $i$ is zero-satisfied at $x'$, and $S' = S$, if $i$ is boundary-satisfied at $x'$.

**Action at Bad Events.** In case of a bad event, note that the coordinate $j$ responsible for the condition in the bad event must belong to $S \cup \{i\}$ because in all other coordinates $(D_S^i(x'))_j = 0$ by definition. Our action depends on which $j$ triggers the bed event as follows:
(1) if the triggering $j = i$, then we start epoch $(i-1, S \setminus \{i-1\})$ at $x'$, otherwise
(2) if the triggering $j \neq i$, then we start epoch $(i, S \setminus \{j\})$ at $x'$.

**Action at Middling Events.** In this case, we start epoch $(i, S \cup \{j\})$ at $x'$ because the coordinate $j$ is both zero- and boundary-satisfied at $x'$ so we add $j$ to $S$ to keep it zero-satisfied next.

Combining the above rules we get a full description of our algorithm in Dynamics 2. In Appendix B we do a step-by-step execution of this algorithm for a simple 2D min-max optimization problem.

---

**Dynamics 2** STay-ON-the-Ridge (STON'R)

---
1: Initially $x(0) \leftarrow (0, \ldots, 0)$, $i \leftarrow 1$, $S \leftarrow \emptyset$, $t \leftarrow 0$.
2: **while** $x(t)$ is not a VI solution **do**
3:     Initialize epoch $(i, S)$'s continuous-time dynamics, $\dot{z}(\tau) = D_S^i(z(\tau))$, at $z(0) = x(t)$.
4:     **while** $z(\tau)$ is not an exit point as per Definition 3 **do**
5:        Execute $\dot{z}(\tau) = D_S^i(z(\tau))$ forward in time.
6:     **end while**
7:     Set $x(t + \tau) = z(\tau)$ for all $\tau \in [0, \tau_{\mathrm{exit}}]$ *(where $\tau_{\mathrm{exit}}$ is the time $z(\tau)$ became an exit point).*
8:     **if** $x(t + \tau_{\mathrm{exit}})$ is (Good Exit Point) as in Definition 3 **then**
9:        **if** $i$ is zero-satisfied at $x(t + \tau_{\mathrm{exit}})$ **then**
10:          Update $S \leftarrow S \cup \{i\}$.
11:        **end if**
12:        Update $i \leftarrow i + 1$.
13:     **else if** $x(t + \tau_{\mathrm{exit}})$ is a (Bad Exit Point) as in Definition 3 for $j = i$ **then**
14:        Update $i \leftarrow i - 1$ and $S \leftarrow S \setminus \{i-1\}$.
15:     **else if** $x(t + \tau_{\mathrm{exit}})$ is a (Bad Exit Point) as in Definition 3 for $j \neq i$ **then**
16:        Update $S \leftarrow S \setminus \{j\}$.
17:     **else if** $x(t + \tau_{\mathrm{exit}})$ is a (Middling Exit Point) as in Definition 3 for $j < i$ **then**
18:        Update $S \leftarrow S \cup \{j\}$.
19:     **end if**
20:     Set $t \leftarrow t + \tau_{\mathrm{exit}}$.
21: **end while**
22: **return** $x(t)$

---

## 5.2 Our Assumptions and Our Main Theorem

We next present the assumptions on $V$ that are needed for our convergence proof. We discuss these assumptions further in Appendix A where we present some high level reasons why they are mild.

**Assumption 1.** *There exist real numbers $\sigma_{\max} > \sigma_{\min} > 0$ such that: for all $x \in [0,1]^n$ and for all $S = \{s_1, \ldots, s_m\} \subseteq [n]$, if $V_\ell(x) = 0$ for all $\ell \in S$, then the singular values of the $m \times m$ matrix $J_S^K(x)$ are greater than $\sigma_{min}$ and less than $\sigma_{max}$, where*

$$J_S^K(x) := \begin{pmatrix} \frac{\partial V_{s_1}(x)}{\partial x_{s_1}} & \cdots & \frac{\partial V_{s_1}(x)}{\partial x_{s_m}} \\ \vdots & & \vdots \\ \frac{\partial V_{s_m}(x)}{\partial x_{s_1}} & \cdots & \frac{\partial V_{s_m}(x)}{\partial x_{s_m}} \end{pmatrix}.$$

Assumption 1 ensures that the direction $D_S^i(\cdot)$ of Definition 2 is uniquely defined (see Lemma 11 in Appendix 8).

**Assumption 2.** *For all $x \in [0,1]^n$, for all $i \in [n]$, and for all $S \subseteq [i-1]$: if $V_\ell(x) = 0 \; \forall \ell \in S$ and $x_\ell \in \{0,1\} \; \forall \ell \notin S \cup \{i\}$ then there is at most one coordinate $j \in S \cup \{i\}$ such that $x_j \in \{0,1\}$.*

Assumption 2 ensures that any time at most one coordinate can trigger a middling or a bad event. To see this, imagine there are two different coordinates $j_1, j_2$ triggering a bad event at $x$, then $x_{j_1} \in \{0,1\}, x_{j_2} \in \{0,1\}$ and $V_{j_1}(x) = V_{j_2}(x) = 0$ and therefore Assumption 2 is violated. A similar observation applies for middling events. See also Lemma 8 and Lemma 10 in the Appendix.

**Assumption 3.** *For all $x \in [0,1]^n$, for all $i \in [n]$, for all $S \subseteq [i-1]$ such that $V_\ell(x) = 0 \; \forall \ell \in S$ and $x_\ell \in \{0,1\} \; \forall \ell \notin S \cup \{i\}$, and for all vectors $(d_{s_1}, \ldots, d_{s_m}, d_i)$ satisfying the equations,*

$$\nabla_{S \cup \{i\}} V_j(x)^\top \cdot (d_{s_1}, \ldots, d_{s_m}, d_i) = 0 \text{ for all } j \in S,$$

*we have that $d_j \neq 0$ if $x_j = 0$ or $x_j = 1$.*

Assumption 3 ensures that we can determine whether a coordinate begins or stops being satisfied by looking at the Jacobian of $V$. For example, consider a coordinate $j$ such that $x_j = 0$ and $V_j(x) = 0$. If also $D_S^i(x)^\top V_j(x) = 0$ then higher-order information is needed in order to determine whether the direction $D_S^i(\cdot)$ makes the coordinate $j$ satisfied or unsatisfied (see Lemma 4 in the Appendix).

We are now ready to state our main theorem.

**Theorem 1.** *Under Assumptions 1, 2, and 3, there exists some $\bar{T} = \bar{T}(\sigma_{\min}, \sigma_{\max}, n, L, \Lambda) > 0$ such that* STAY-ON-THE-RIDGE *(Dynamics 2) will stop, at some time $T \leq \bar{T}$, at some point $x(T) \in [0,1]^n$ that is a solution of* $\mathrm{VI}(V, [0,1]^n)$.

**Remark 1** (Discrete-time Algorithm). *It is possible to combine the proof of Theorem 1 with standard numerical analysis techniques to show the convergence of a simple discrete version of the dynamics assuming that the step size is small enough. For more details about this we refer to Appendix J.*

## 5.3 Simulated 2-Dimensional Example

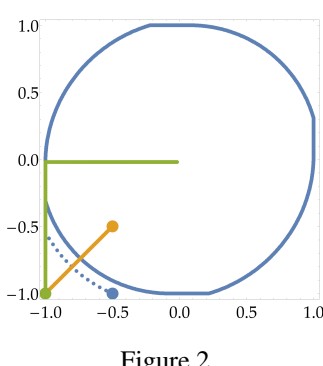

Figure 2

In Figure 2 we present the behavior of the main existing algorithms for min-max optimization in the 2-dimensional min-max problem with objective $f(\theta, \omega) := -\theta\omega - \frac{1}{20} \cdot \omega^2 + \frac{2}{20} \cdot S\left(\frac{\theta^2 + \omega^2}{2}\right) \cdot \omega^2$, where $S(z)$ is the smooth-step function equal to 0 for $z \leq 0$, 1 for $z \geq 1$ and $z^2 - 2z^3$ otherwise. With **blue** we observe the behavior of GDA, EG, and OGDA that have the same behavior in this example when initialized at $(-0.5, -1)$. With **orange** we observe the behavior of the follow-the-ridge (FtR) algorithm initialized at $(-0.5, -0.5)$ and with **green** we observe the behavior of STON'R. As we can see GDA, EG, OGDA are getting trapped to a cycle whereas FtR hits the boundary at $(-1, -1)$ that does not correspond to an equilibrium point. Our algorithm is the only one that directly converges to the equilibrium following a very short path. In Appendix C we provide a more detailed explanation of this example and we observe similar behavior for different initializations of GDA, EG, OGDA, and FtR.

### 5.4 SKETCH OF PROOF OF THEOREM 1

For a sketch of our proof of Theorem 1 we follow the recipe that we described in Section 4. During this proof sketch we highlight some technical challenges that we face. (The full proof can be found in Appendix E.)

1. We start with the definition of the set of nodes $N$. The set $N$ contains triples of the form $(i, S, x)$ where $i \in [n]$, $S$ is a subset of $[i-1]$ and $x \in [0, 1]^n$ that satisfies the following:

   (a) all coordinates in $S$ are zero-satisfied, (b) all coordinates in $[i-1] \setminus S$ are boundary-satisfied, (c) $x_j = 0$ for all $j \geq i + 1$, and either (d1) $x_i = 0$ or (d2) $x$ is an exit point for epoch $(i, S)$ according to Definition 3 [1].

   Our first technical challenge is to show that the size of $N$ is finite (see Lemma 3 in the Appendix). Next we describe a mapping $F : N \to N$. Let $(i, S, x) \in N$, we use the dynamics $\dot{z} = D_S^i(z)$ with initial condition $z(0) = x$ and we find the minimum time $\tau_{\text{exit}}$ such that $z(\tau_{\text{exit}})$ is an exit point. We then update $i, S$ to $i', S'$ according to the rules for actions on exit points of Section 5.1 and we define $F((i, S, x)) = (i', S', z(\tau_{\text{exit}}))$. One of our main technical challenges is to show that the dynamics $\dot{z} = D_S^i(z)$ have a unique solution under our assumptions and hence $F$ is well defined (see Lemma 4 in the Appendix).

   The set $N$ and the mapping $F$ define the directed graph $G$, as described in Section 4, that is guaranteed to have vertices with out-degree at most 1. We also show that any $v \in V$ with out-degree 0 is an equilibrium point (see Lemma 4 in the Appendix).

2. To show that the in-degree is at most 1, we face our next technical challenge which is to show that we can actually solve the dynamics backwards in time. In particular, if we specify $z(0)$ and there is the smallest time $\tau_{\text{exit}}$ such that $z(-\tau_{\text{exit}})$ is an exit point then $z(-\tau_{\text{exit}})$ is uniquely determined. This means that there exists $F^{-1} : N \to N$ such that if $v' = F(v)$ then $F^{-1}(v') = v$ which means that no vertex in $N$ can have in-degree more than 1 (see Lemma 5 in the Appendix).

3. We show that $v_0 = (1, \emptyset, (0, \ldots, 0)) \in N$ and that if run the dynamics $\dot{z} = D_\emptyset^1(z)$ backwards in time starting at $z(0) = 0$ then we get outside $[0, 1]^n$ and so $v_0$ has in-degree 0. We also show that the dynamics $\dot{z} = D_\emptyset^1(z)$ can move forward in time and stay inside $[0, 1]^n$ so $v_0$ has out-degree 1 (see Lemma 6 in the Appendix).

4. The above show that our algorithm converges according to Section 4.

## 6 CONCLUSIONS

**Summary.** In this work we propose a novel local-search algorithm, called STON'R, that is guaranteed to converge to local min-max equilibrium in the general case of non-convex non-concave objectives. To the best of our knowledge STON'R is the first method, beyond trivial brute-force, that is guaranteed to find a local min-max equilibrium starting from a simple initialization. We remark that existing min-max optimization methods required either convexity (resp. concavity) in one of the players or an initialization very close to the optimal point in order to guarantee convergence. Finally, our approach differs from existing methods in the fundamental way that both its design and analysis are based on topological rather than potential arguments. We believe that these types of arguments can play an important role in the future of multi-agent machine learning.

**Comparison with Brute-Force.** Since we assume that $V$ is a Lipschitz function and that $K$ is an $n$-dimensional hypercube, it is not hard to see that there exists a small enough discretization of the space such that the brute-force search over all the discrete points is guaranteed to find a solution. Such brute-force algorithms exist in most of the optimization problems like solving linear programs or finding Nash equilibria in normal form games. These trivial algorithms suffer from the curse of dimensionality even in very simple instances and hence they are almost never useful. Instead local-search algorithms such as simplex or Lemke-Howson Lemke & Howson (1964) have been extremely successful in practice because they converge very fast in the majority of real world instances although in the worst-case their complexity is the same as the brute-force. Our contribution is to provide the first such algorithm for the fundamental problem of nonconvex-nonconcave min-max optimization and we believe that it will play an important role in the future of multi-agent optimization in machine learning.

---

[1] The actual set of nodes that we used in the proof does not contain the information of $i$ and $S$ but we refer to the Appendix for the exact proof.

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
