# OpenReview forum: "STay-On-the-Ridge (STON'R): Guaranteed Convergence to Local Minimax Equilibrium in Nonconvex-Nonconcave Games"
_ICLR.cc/2023/Conference — Submitted to ICLR 2023_

### Official Review · Reviewer_5vqs · 2022-10-21

**Confidence:** 4
**Correctness:** 2
**Technical Novelty And Significance:** 3
**Empirical Novelty And Significance:** Not applicable
**Recommendation:** 5

**Clarity, Quality, Novelty And Reproducibility:**

Clarity is the major problem of the paper, which also poses question marks regarding correctness.

A few minor comments:

- Is there any reason for considering min-max optimization instead of general $k$-player games? The argument seems to work for general variational inequality problems.

- In the proof of Lemma 9, "...for an appropriately selected $L$": $L \rightarrow M$.

- In the definition of $F^i(x)$ at page 21, what does it mean that "coordinate $\ell$ is fixed at $x$? It seems like the authors are suggesting that $F^{i}(x)$ are the set of points that are boundary satisfied. (Does it mean "frozen" as in Definition 6?)


**Strength And Weaknesses:**

Strength:

- The ideas are substantially new.

Weaknesses: While I really appreciate the highly innovative algorithm and ideas, I feel like the paper, in its current state, is more of a draft than is ready to be published. I have three main concerns:

1. The presentation, especially on the proof, is very messy and difficult to follow. Here are only a few examples:
   - The argument for "Assumption 1 is mild" is very vague. For instance, can the authors explain why adding a periodic function wouldn't change the VI solutions? In its current form, the authors only gave an explanation to approximate stationary points but nothing on the boundary. Moreover, what is the basis for "In higher dimensions we do not provide a formal argument for a reduction, although we conjecture that it is true."?
   - The authors argued that Assumption 2 is mild since one can start in a smaller (random) subset of $[0,1]$. In this case, where do we initialize STON'R? Does this not destroy the pivot argument in the proof for Theorem 1? Moreover, the curve $C$ in Example 1 of Section C.2 is just a singleton $(1,0)$ so I don't really grasp its meaning.
    - Why does Lemma 8 imply Lemma 3? In Definition 7, pivots include boundary satisfied points, whereas Lemma 8 only deals with the cardinality of 0-satisfied points.

    This list can go on. Essentially, whenever I check for a rigorous statement, there is some ambiguity. I believe many of them can be easily fixed, but in its current form I find it almost impossible to verify the correctness of the paper. Since this paper is all about theory, I consider it a serious flaw.

2. The relevance of the proposed algorithm to machine learning is unclear:
    -  The authors claimed that the proposed algorithm is second-order, which is not really true. At best, it can be described as a **piecewise** second-order algorithm involving various breakpoints, but it is significantly more complicated than any bona fide second-order method. In any case, it is computationally much more demanding than existing second-order algorithms such as (Wang et al., 2019).

    - The claim that the authors have solved the open problem "Is there an algorithm which is guaranteed to converge to a local min-max equilibrium in the nonconvex-nonconcave setting (Wang et al., 2019)?" is an overstatement for me. Since the authors **changed** the problem formulation to constrained min-max problems, a simple grid search + Follow-the-Ridge of (Wang et al., 2019) would work; note that this naive algorithm is more practical than STON'R.

        This is not intended to say that the contribution of the paper is trivial; it simply points out that the interest of the paper mainly lies in its theoretical insights, something that is not accessible from prior works. I therefore suggest the authors to rephrase their contribution.


3. The argument seems to be highly specific for box constraints. For instance, how do I extend the algorithm to $\ell_2$-balls?

In addition, there are some issues with the reduction to VIs in Section 2. For instance, a local **max-min** (instead of min-max) in the interior would verify the VI proposed at the bottom of page 3, contradicting the authors' claim that VIs are equivalent to local min-max. Can the authors clarify?



**Summary Of The Paper:**

This submission proposes a new algorithm called the STay-ON-the-Ridge (STON'R) for converging to the local minimax of constrained nonconvex-nonconcave games. The problem is reformulated in terms of variational inequalities. The key idea is to find a set of points on which one can define a directed graph with in- and out-degree at most one, with one node being the sought equilibrium.

**Summary Of The Review:**

The paper proposes contains many novel ideas. However, my concerns are that
1. The presentation is messy and it is almost impossible to check the proofs.
2. Stylistically speaking, I believe this is more of a TCS paper than an ML paper. The relevance to ML is not very clear.

---

> ### Author Response · Authors · 2022-11-10
> **Response to Reviewer 5vqs - Part 3**
>
> ## 3.
>
> > "The argument ..."
>
> (a) The box constraints are a very fundamental instance of the problem and even for this fundamental instance, prior to our work there was no local-search method with guaranteed convergence.
>
> (b) Our technical part of the paper is using the formulation of variational inequalities over convex compact sets K. If we assume that there exists an 1-to-1 continuous map $G_K : K \to [0, 1]^n$ that is sufficiently smooth then instead of solving the problem $VI(V, K)$ we can compute a solution $\tilde{x}$ of the problem $VI(V \circ G_K^{-1}, [0, 1]^n)$ and then $G_K(\tilde{x})$ should be solution of $VI(V, K)$. This is a standard type of argument in many fixed point computation problems, which are equivalent with VIs when K is a convex compact set. Nevertheless, making this precise requires many additional definitions and techniques that would make our paper more difficult to read. So we decided not to mention it since the hypercube constraints $[0, 1]^n$ are very important on their own.
>
> > "In addition,...clarify?"
>
> The $(\epsilon, \delta)$-local min-max equilibria when $\delta \le \sqrt{\Lambda/\epsilon}$ are equivalent to fixed points of the project GDA dynamics as was proved in Theorem 5.1 of Daskalakis et al. 2020. So when we are looking for a solution in the interior, any point with zero gradient will be a solution, in particular a local max-min or local max or local min. In other words we consider first-order local min-max equilibria. In the general and difficult setting that we consider though:
> (a) This is the only type of local equilibria that is guaranteed to always exist. Second-order local min-max equilibria might exist or might not exist.
> (b) The problem of computing first-order local min-max equilibria in constrained problems is already exponentially difficult. see Daskalakis et al. 2021.
> (c) In many instantiation of even single objective nonconvex optimization, a natural goal is to compute first-order stationary points, although they might correspond to saddle points instead of local optima.
>
> ## Response to Minor Comments
> + The reviewer is right that STORN can be applied to general variational inequality problems.
> + We fixed the typo.
> + The reviewer is right.  With the phrase “Coordinate $\ell$ is fixed” we meant “Coordinate $\ell$ is boundary satisfied”. We fixed the typo.

---

> ### Author Response · Authors · 2022-11-10
> **Response to Reviewer 5vqs - Part 2**
>
> ## 2.
> > "...The authors claimed that the proposed algorithm is second-order,...(Wang et al., 2019)."
>
> The term “second-order algorithm” can maybe have several meanings in different groups. What we mean with second-order is that our algorithm is an iterative local-search algorithm and the only access it has to the objective function is via its first and second derivatives. What definition of second-order algorithms do you have in mind that does not capture our method? We do not understand how the existence of breakpoints can decrease the value of our algorithm for the ML community.
>
> The algorithm of Wang et al., 2019 needs time $n^3$ at every iteration to invert the Hessian whereas we need $n^3$ to solve a linear system that involves the Hessian. Why is our algorithm much more demanding? If anything, solving linear systems can be slightly easier in practice than inverting matrices. Moreover, notice that Follow-the-Ridge is guaranteed to converge only in case it is initialized near the local min/max solution. This is not just a theoretical issue since Follow-the-Ridge diverges even in simple 2D examples (see Appendix C).
>
> > "The claim...contribution."
>
> Indeed, since we assume that $V$ is a Lipschitz function and that $K$ is an $n$-dimensional hypercube, there exists a small enough discretization of the space such that the brute-force search over all the discrete points is guaranteed to find a solution. Such brute-force algorithms exist in most of the known optimization problems like solving linear programs or finding Nash equilibria in normal form games. These trivial algorithms, though, suffer from the curse of dimensionality even in very simple instances and hence they are almost never useful. Instead local-search algorithms such as simplex or Lemke-Howson have been extremely successful in practice because they converge very fast in the majority of real world instances although in the worst-case their complexity is the same as the brute-force. Our contribution is exactly to provide the first such algorithm for the fundamental problem of nonconvex-nonconcave min-max optimization and we believe that it will play an important role in the future of multi-agent optimization in machine learning.
>
> Of course, naive implementations of such local-search algorithms will not directly be efficient in practice and a significant effort is needed to end-up with programs that can efficiently be applied in large real-world instances. Our contribution is to prove that such a local-search algorithm with guaranteed convergence exists for nonconvex-nonconcave min-max optimization and we believe that efficient implementations of this algorithm can have many applications in multi-agent ML.
>
> We changed our wording in the introduction and we added part of the above discussion in the conclusions section of our revision.
>
> > "...a simple grid search...would work;"
>
> As we mentioned, simple grid search (brute-force) is sufficient, there is no need to apply Follow-the-Ridge as well and there is no quantitative improvement from using it.
>
> > "note that this...than STON'R."
>
> Why would simple grid search + Follow-the-Ridge be a more practical method than STON’R?
> Our 2-dimensional examples in Appendix A (Appendix C in the revision) show the opposite actually. There are simple 2D examples (see Figure 6 (b)) where FtR diverges for almost all initializations on a fine grid. And hence any algorithm that involves grid search and FtR would query almost every point in the grid. In contrast STON’R follows a very short path and asks a very limited number of queries until it finds the solution.

---

> ### Author Response · Authors · 2022-11-10
> **Response to Reviewer 5vqs - Part 1**
>
> We thank the reviewer for reading our paper carefully, for appreciating the novelty in our results and for their thoughtful comments!
>
> ## 1.
> > "The presentation,...follow."
>
> In the updated version of the paper we have substantially restructured the proof of our main Theorem 1. We would gladly hear the comments of the reviewer for the new presentation and we are more than willing to answer/clarify further questions related to the proofs as well as incorporate changes/suggestions that would result in better presentation.
>
> The reasoning that we provide for why Assumptions 1 - 3 are mild is certainly not a complete formal proof and this was our intent from the beginning. Although we conjecture that we can indeed get rid of these assumptions if we had a complete formal proof we would not mention these assumptions in our main result. We changed our wording in Section 5.2 to make it clear that we only provide high-level reasoning and not formal arguments. Below we clarify the **high-level** reasons that we believe that Assumptions 1 - 3 are mild.
>
> > "The argument for "Assumption 1 is mild" is very vague. ..."
>
> We updated this section in the Appendix of our revision. Once a perturbation $\tilde{V}$ with norm at most $\epsilon$ is added to the original objective function then it is immediate that any $\alpha$-approximate solution of perturbed VI is an $(\alpha + \epsilon \cdot R)$-approximate solution of original VI, where $R$ is the diameter of the domain K (see Section A.1). This is a fact that can be easily checked via algebraic manipulations which we added in our revision.
>
> > "Moreover, what is the basis..."
>
> Our conjecture is that there exists an appropriate perturbation $\tilde{V}$ in high dimensions that allows us to reduce a function where Assumption 1 is false to a function where Assumption 1 is true. Simple perturbations such as periodic functions over random $n$-dimensional lattices seem to work in some high-dimensional examples that we tried, but we do not have a proof.
>
> Again, we never said that we have complete formal proofs for why Assumptions 1 - 3 are mild, we just have some high-level reasoning.
>
> Notice also that a very similar assumption on the singular values of the Hessian appears in the recent work “Last-Iterate Convergence Rates for Min-Max Optimization: Convergence of Hamiltonian Gradient Descent and Consensus Optimization”.
>
> > "The authors...meaning."
>
> Any set of the form $[a_1 ,b_1] \times \ldots \times [a_n ,b_n]$ can be transformed to the $[0,1]^n$ by considering the variable $z_i = (x_i - a_i) / (b_i - a_i)$. Then STON’R can be applied for the $[0,1]^n$ hypercube with respect to the z-variables and can be initialized at $(0,...,0)$ as we mention in our revision.
>
> The purpose of the example was to illustrate how Assumption 2 can be ensured with an $\epsilon$-perturbation on the boundary.  Let the function $f(x,y) = -x^2/ + yx - x$ where $x$ is the min and $y$ is the max. The above function $f$ does not satisfy Assumption 2 since point (1,0) coordinate x is zero-satisfied and coordinate y is on the boundary. Notice that the x coordinate is zero-satisfied only on the curve $C:= \{ x - y = 1 \} \cap [0,1]^2$ (the curve in the current example). As Example 1 shows after $\epsilon$-permutation on the boundary, there is 0 probability that x is zero-satisfied and both coordinates (x,y) belong in the boundary, meaning that Assumption 2 is satisfied.
>
> We agree with the reviewer that since  $C:= \{ x - y = 1 \} \cap [0,1]^2$ is a singleton, the above example may seem a bit weird. We have replaced it with the similar example $f(x,y) = -x^2/ + y^2 + y$ that again does not satisfy Assumption~2 because of the point $(0,0)$. Notice that coordinate x is zero satisfied only along the curve $C:= \{ x - y = 0 \} \cap [0,1]^2$ and thus (as in the current example) after an $\epsilon$-perturbation on the boundaries there is $0$ probability one of the corners of $[0,1]^2$ lies in C.
>
> > "Why does Lemma 8...points."
>
> By Definition 7 a pivot admits $m$ coordinates on the boundary and $n-m$ coordinates that are zero-satisfied. By fixing a specific set of coordinates to be on the boundary, Lemma 8 establishes that there is a finite set of points at which the rest of the coordinates are zero-satisfied. Since there are at most $2^n$ possible values for the boundary coordinates, the overall number of pivots is finite. We have added the above discussion in the updated version of the paper. You can find it in after the proof of Lemma 7 (the previous Lemma 8).
>
> > "This list..."
>
> As we explained above it was never our intent to provide formal proof for why Assumptions 1 - 3 are mild, otherwise we wouldn’t have included them in our main theorem. The implication of Lemma 3 from Lemma 8 is straightforward and we updated our manuscript. We do not believe there are flaws in the proof of Theorem 1 and we are more than happy to add any clarifications in specific places where our proof is not clear.

---

### Official Review · Reviewer_PZ8h · 2022-10-24

**Confidence:** 3
**Correctness:** 3
**Technical Novelty And Significance:** 3
**Empirical Novelty And Significance:** 3
**Recommendation:** 5

**Clarity, Quality, Novelty And Reproducibility:**

**Clarity:**

This paper is generally clear. There are a few points that need to be clarified as listed below.

(1) Right after eq. (1), you said "A solution to (1) corresponds to a Nash equilibrium of this sequential-move game." Could you give the definition? Why does "Nash equilibrium of the simultaneous-move game also constitutes a Nash equilibrium of the sequential-move game"? Could you give a brief proof or a citation that contains proof?

(2) At the beginning of Section 2, in the definition of VI, should it be $V(x)^{\top}\cdot(x-y)\ge 0$ instead of $\le 0$?

(3) Why do we need condition 3 of Definition 2? You may add the explanation after Definition 2.

(4) In Assumptions 2 and 3, why not let $i>\max(s_1,\ldots,s_m)$ and $x_{\ell}\in${$0,1$} for only $\ell\in [i-1]\backslash S$ as seemingly required by the algorithm? Could you also add brief and intuitive explanation on why Assumptions 2 and 3 ensures unique $j$ in bad and middling events?

(5) You mentioned in Section 5.1 that Assumption 1 guarantees unique direction $D_S^i(x)$ and that Assumptions 2 and 3 guarantee unique $j$. Could you cite the Lemmas that prove the uniqueness?

(6) In Appendix A, What are the coordinates of local min-max equilibria for $f_1$ and $f_2$?


**Quality:**

The convergence looks well supported by both theorem and experiments.


**Novelty:**

This paper is highly novel in at least 2 aspects: First, this work proposes the first algorithm that is guaranteed to converge to a local min-max equilibrium for smooth nonconvex-nonconcave minimax optimization. Second, the topological proof of convergence is elaborate and not commonly used.


**Reproducibility:**

Hyperparameters such as stepsizes of GDA and EG could be given to ensure reproducibility of experiments.


**Minor comment:**

At the beginning of page 5, the two comments "this is so that XX is maintained" could be expressed as "this guarantees XX" or "To guarantee XX".

**Strength And Weaknesses:**

Pros: This work is highly novel in both algorithm and proof technique, as elaborated in **Novelty** below. The convergence is also well supported by both theorem and experiments.

Cons: A few points need to be clarified as listed in **Clarity** below.

**Summary Of The Paper:**

To the authors' knowledge, this work proposes the first algorithm that provably converges to a local min-max equilibrium for smooth nonconvex-nonconcave minimax optimization.

**Summary Of The Review:**

This paper is very well written, which studies the unsolved challenging problem of finding local minimax equilibrium of nonconvex-nonconcave minimax optimization, with highly novel algorithm. The convergence is proved in a novel topological way, and also well demonstrated by experiments. There are a few unclear points which I think are not hard to clarify. Therefore, I would like to see this paper accepted.

---

> ### Author Response · Authors · 2022-11-10
> **Response to Reviewer PZ8h**
>
> We thank the reviewer for reading our paper carefully, for appreciating our results a lot and for their thoughtful comments!
>
> (1) Our paper focuses on the simultaneous-move game and we don’t think we have enough space to define the equilibria of sequential-move games and show their relation with simultaneous-move games. A careful exploration of these notions is given in [1]. See for example Figure 2 of [1]. We added a reference to [1] after the second sentence.
>
> [1] “What is Local Optimality in Nonconvex-Nonconcave Minimax Optimization?” Chi Jin, Praneeth Netrapalli, Michael I. Jordan
>
> (2) We fixed the typo.
>
> (3) Condition 3 is very important in Definition 2. Conditions 1 and 2 specify a line in the n-dimensional space, whereas Condition 3 gives a rule to choose one of the two directions on this line. We comment on that in the second bullet of Section 3 and we added a comment for this right after Definition 2 in our updated version.
>
> (4-5) We have added in Section 5.2 below the statements of the Assumptions 1 - 3 an explanation of the importance of each assumption and we also added links to the corresponding lemmas in the Appendix.
>
> (4 cont’d) Yes you are right, thanks for the suggestion. We were stating a slightly stronger assumption than what we needed, we updated our Assumptions 2 and 3 in our revision.
> Let us know if everything is clear now.
>
> (6) In the experimental part of the Appendix, for both of the functions f1 and f2, the only local min-max equilibrium is at (0,0) assuming that the constraint set is $ [-1,1]^2$.
>
> > "Minor comment: ..."
>
> Thank you for the suggestion we fixed it in the revision!

---

> > ### Comment · Reviewer_PZ8h · 2022-11-18
> > **2nd reply from Reviewer PZ8h**
> >
> > I'm satisfied with authors' reply and revision and keep 8.

---

### Official Review · Reviewer_Ej7m · 2022-10-28

**Confidence:** 3
**Correctness:** 4
**Technical Novelty And Significance:** 4
**Empirical Novelty And Significance:** 3
**Recommendation:** 5

**Clarity, Quality, Novelty And Reproducibility:**

Please see the above review for further details.


**Strength And Weaknesses:**

The paper proposes a novel idea for the convergence to a local min-max equilibrium for smooth nonconvex-nonconcave objectives. The fact, that the proposed method and the analysis are motivated by the topological nature of the problem positions this paper in a different category than other classical and recent works.  As the authors mentioned the method is not designed to decrease some potential function but is designed to satisfy a topological property that guarantees the avoidance of cycles and implies its convergence.

Having in mind the tight timeline for the submission of the reviews (less than 2 weeks to review 3-5 papers), I would be positively surprised if any of the reviewers were able to follow and understand the proof techniques of this work. The paper is heavily theoretical and the proof arguments are not standard.

Main Issue: Presentation

As it is now the paper gives the impression that it was rapidly cut before the submission to simply fit into 9 pages (looks more like a draft of a final paper). I understand that the authors try to fit the novel idea into the given ICLR space but in my opinion, this is an impossible task for such results. For having a proper presentation and allowing the reader to appreciate every aspect of this work I believe one needs much more space.

Some suggestions for squeezing the results but keeping important parts in the main paper:

1. I would suggest removing section 5.3 in the appendix and adding some experiments in the main paper as well as a conclusion.
2. In part of the main contributions in section 1 a table with what are the existing results in terms of convergence guarantees and the main contribution will give a better idea to the reader of what is needed. A paragraph explaining what are the challenges of the new approach will be also needed there.

Question:

How this method is related to the Hamiltonian gradient and consensus optimization methods proposed in Abernethy et al. (2019)? Why these second-order methods cannot be used in the setting under study?

Some missing references from related work on last-iterate convergence:

[1] E. Gorbunov, N. Loizou, and G. Gidel. Extragradient method: O(1/K) last-iterate convergence for
monotone variational inequalities and connections with cocoercivity. AISTATS 2022

[2] Yang Cai, Argyris Oikonomou, Weiqiang Zheng
Finite-Time Last-Iterate Convergence for Learning in Multi-Player Games, NeurIPS 2022

**Summary Of The Paper:**

The paper proposes Stay-On-the-Ridge $(STON'R)$ algorithm, which according to the authors it is the first method that is guaranteed to converge to a local min-max equilibrium for smooth nonconvex-nonconcave objectives. The proposed method is a second-order algorithm that provably escapes limit cycles as long as it is initialized at an easy-to-find initial point. Finally, the algorithm is designed to satisfy a topological property that guarantees the avoidance of cycles and implies its convergence.

**Summary Of The Review:**

The paper in its current format is very hard to read.

I believe the approach and results are interesting however a more clear presentation will be needed for the results to be able to be understood and get appreciated by a broader audience.

For this reason, I gave a score of "5: marginally below the acceptance threshold" for this work.

---

> ### Author Response · Authors · 2022-11-10
> **Response to Reviewer Ej7m**
>
> We thank the reviewer for reading our paper carefully, for appreciating the novelty in our results and for their thoughtful comments!
>
> > "Having in...standard.”
>
> We agree that the reviewing time was not sufficient but we believe that this should not reflect decisions on papers that introduce a lot of new ideas.
>
> > "Main Issue: Presentation ..."
>
> Indeed several papers, especially theoretical ones, and those with many new ideas might need more than 9 pages for the full proofs to be presented. It is thus standard that the full proof details are deferred to the appendix. In this community, important works are made public via top-tier conferences such as ICLR and also people that participate in such conferences are expecting to see important new results even if they do not perfectly fit in 9 pages. Given all that, we believe that space limitations and time-constraints in the reviewing process should not exclude novel works from being published. We are aware of several papers in ICLR where the 9-pages contain the highlights of the results  and important technical insights while the complete details are deferred to the appendix. In our updated manuscript we followed reviewer’s suggestions and we believe that our 9-pages, although dense, illustrate the exact statements, our main ideas, the importance and the context in the literature of our result. We are willing to update our manuscript again if the reviewer has further suggestions.
>
> > "Some suggestions ..."
>
> Thank you for your suggestions!
> 1. We managed to add one figure with a simulated experiment as well as conclusions while keeping 5.3 (now 5.4). We believe that the proof sketch is important to be in the main part because it is the place where we can concretely mention what are the technical challenges and highlight the main steps of the proof.
> 2. We added a table at the end of Section 1. We also added a few sentences explaining the challenges in Section 1.1 and in the proof sketch in Section 5.4.
>
> Let us know if you believe that there are more updates that are needed.
>
> > "Question: ..."
>
> Abernathy et al. provide convergence rates for Hamiltonian descent and Consensus Optimization in the following cases i) strongly convex / strongly convex ii) non convex / linear iii) sufficiently bilinear. Thus, their results do not apply in the general nonconvex-nonconcave setting that we consider. More precisely, their method is not enough even to guarantee avoidance of cycles in our general nonconvex-nonconcave setting.
>
> > "Some missing references ..."
>
> Thank you for the references, we included them in our revision.

---

### Decision · Program_Chairs · 2023-01-20

**Decision:**

Reject

**Justification For Why Not Higher Score:**

This was not an easy decision because of the potential in the paper, but eventually the negatives outweighed the positives, as explained above.

**Justification For Why Not Lower Score:**

N/A

**Metareview: Summary, Strengths And Weaknesses:**

The presents an algorithm to solve variational inequalities over the unit cube under some seemingly mild assumptions. This implies that the algorithm can be used to find approximate local Nash equilibrium points for min-max optimization problems with non-convex-non-concave smooth objectives.

The reviewers agree that the paper proposes an interesting and novel algorithm, and I think that the main ideas are nicely flashed out in the main body of the paper. However, the reviewers found that the results are not stated properly and also that it was hard to verify the proofs. Although the authors made changes during the revision, it was not possible to fully check all the changes. The paper definitely has more typos left, e.g., the variational inequalities are still reversed in the proof of Lemma 1, although used correctly, or the definition of a pivot point should either state that there is an unsatisfied variable or $\ell$ should be defined to be at least $n$ in case no such variable exists (or make the same assumption in Lemma 3 instead), or Lemma 8 is referenced in place of Lemma 7 when proving Lemma 3, etc. While these issues are not serious and can be easily corrected, given the previous concerns of the reviewers, they suggest that the paper could benefit from another review cycle.

More importantly, a revision should clarify the contributions better. Regarding the main results concerning variational inequalities, it would be good to quantify the dependence of $\bar{T}$ on its parameters in Theorem 1, and that of $\bar{M}$ in Theorem 2, together with some intuition (formula) how small the step size should be. Getting stronger arguments about why the assumptions are mild would also substantially improve the results. With respect to the approximate local Nash equilibrium, it would be important to clarify the meaning and limitation of this concept. As the authors also explained in their response, the approximate local Nash equilibrium concept is quite "fragile", as, for smooth functions, it is satisfied by any stationary point. Hence, one can easily find such a point by running gradient descent over both parameters (with sufficiently small step size). While this is clearly not an intended solution, the discussion of such issues would be needed.

Based on the above, unfortunately I cannot recommend acceptance of this paper, but I am convinced that when properly revised, this might become an important contribution to the field. Personally, I would present the main contribution (title, introduction, etc.) about the results on variational inequalities, and then present min-max optimization (or even more generally, multi-player games) as an important application of the method.

Finally, the authors might consider to compare their method to the SGA algorithm of Balduzzi et al., The Mechanics of n-Player Differentiable Games, ICML 2018, and also describe how they parametrized (e.g., discretized for STON'R) the methods presented.